# Vitamin D Merging into Immune System-Skeletal Muscle Network: Effects on Human Health

## Clara Crescioli

Department of Movement, Human and Health Sciences, Section of Health Sciences, University of Rome "Foro Italico", Piazza L. de Bosis 6, 00135 Rome, Italy; clara.crescioli@uniroma4.it; Tel./Fax: +39-06-36733395

**Abstract:** The concept that extra-skeletal functions of vitamin D impact on human health have taken place since quite ago. Among all, the beneficial effects of vitamin D on immune regulation, skeletal muscle function, and metabolism are undeniable. Adequate vitamin D levels maintain the immune system and skeletal muscle metabolism integrity, promoting whole-body homeostasis; hypovitaminosis D associates with the important decline of both tissues and promotes chronic inflammation, which is recognized to underlie several disease developments. Growing evidence shows that the immune system and skeletal muscle reciprocally dialogue, modulating each other's function. Within this crosstalk, vitamin D seems able to integrate and converge some biomolecular signaling towards anti-inflammatory protective effects. Thus, vitamin D regulation appears even more critical at the immune system-muscle signaling intersection, rather than at the single tissue level, opening to wider/newer opportunities in clinical applications to improve health. This paper aims to focus on the immune system-skeletal muscle interplay as a multifaceted target for vitamin D in health and disease after recalling the main regulatory functions of vitamin D on those systems, separately. Some myokines, particularly relevant within the immune system/skeletal muscle/vitamin D networking, are discussed. Since vitamin D supplementation potentially offers the opportunity to maintain health, comments on this issue, still under debate, are included.

**Keywords:** vitamin D; VDR; immune system; skeletal muscle; myokines; inflammation

---

## 1. Introduction

Evidence on the importance of vitamin Ds' extra-skeletal effects on human health is accumulating. Vitamin D is an evolutionary, very old molecule produced in the skin by UV-B radiation-induced photochemical/thermal conversion of the cholesterol precursor 7-dehydrocholesterol [1]. Two enzymatic reactions in the liver (by 25-hydroxylase, CYP24A1) and in proximal tubule renal cells (by 1$\alpha$-hydroxylase, CYP27B1), respectively, convert the inactive precursor first to 25-hydroxyvitamin D3 [25(OH)D], then to the biologically active molecule 1$\alpha$,25-dihydroxyvitamin D3 [1,25(OH)$_2$D]. Upon the binding and interaction with the cognate nuclear vitamin D receptor (VDR), vitamin D can directly regulate the expression or transrepression of several gene products [2]. As studies on the VDR have progressed, vitamin D identity turned from an UV-B absorption product, likely useful for radiation protection, to a multifaceted endocrine molecule playing different physiological effects, as it behaves in higher species [3–5]. Historically, the classical action of vitamin D is the control of bone remodeling and calcium homeostasis. The strong association between vitamin D poor status and bone disorders or infections is well known. Indeed, more than one century ago, sun UV-B exposure or codfish liver oil (rich in vitamin D content) were recommended as treatments to increase vitamin D level for the protection against rickets or tuberculosis [6–8]. From the pioneering studies, and considering the plethora of biological actions, pleiotropic effects of this hormone have emerged [9]. Indeed, a very large number (approximately 250) of tissue and cell types express the VDR gene, suggesting a broad spectrum

of physiological roles unrelated to calcium homeostasis, as shown, i.e., by investigations of gene expression profiling [10]. Importantly, several observational studies evidenced a tight link between low vitamin D and inflammatory status underlying the pathogenesis of some human pathologies, i.e., immune and autoimmune diseases, metabolic disturbances, cardiovascular (CV), and muscular diseases [11–16]. Among all, the deterioration of the immune system and skeletal muscle associated with hypovitaminosis D seems to play a pivotal role in view of the heavy disease burden related to those pathologies. Undeniably, well-functioning immune and muscular systems regulate whole-body homeostasis, and largely contribute to physiological integrity and general good health. Conversely, the dysregulation or loss of skeletal muscle function, i.e., naturally occurring in aging, associates with immune system dysregulation and chronic inflammation, often promoting the development of overlapping diseases [17–21]. In this scenario, growing evidence highlights the importance of the dynamic dialogue existing between the immune system and skeletal muscle, almost behaving like an "immune-muscle axis". This paper aims to focus the attention on the immune system-skeletal muscle bidirectional crosstalk as a target point of vitamin D in health and disease after providing a rapid overview on the critical role of this hormone in the regulation of the immune response and muscular function, separately. The function of some myokines particularly relevant to the network linking the immune system/skeletal muscle/vitamin D signaling, is discussed. Finally, a comment on vitamin D blood determination and supplementation is addressed, considering that these specific issues are hot topics, currently still under debate [22].

## 2. Vitamin D Coordinates the Anti-Inflammatory Immune Response

VDR agonists are known since quite ago as fine-tuned modulators of the immune system. VDR is present in the majority of immune cell types: T cells, including CD4+ and CD8+ subtypes, B cells, neutrophils, and antigen-presenting cells (APCs), as macrophages and dendritic cells (DCs) [23], and it is expressed at different levels, according to the physiological phase and immune signal modulation. During the T cell activation, i.e., a broad VDR upregulation occurs, while the receptor is expressed at very low levels in naïve T cells. Conversely, a significant VDR downregulation associates with macrophages or DCs maturation from monocytes [24–27]. It is well known that vitamin D actively participates to innate immune response building by promoting macrophage chemotactic and phagocytotic activity. This hormone can finely drive immune-adaptive responses as well, by targeting APCs, mainly DCs. Indeed, it counteracts DCs maturation/differentiation from monocytes, inhibits the expression of both costimulatory molecules and the major histocompatibility complex (MHC)-II-complexed antigen on the DC surface, consequently impairing antigen processing and presentation to T cells [28]. In addition, vitamin D directly polarizes T cell-responses from T helper (Th)1/Th17 inflammatory phenotypes to the Th2 tolerogenic subset. Notably, many effects of vitamin D on immune response polarization are driven by a straight inhibition of pro-inflammatory Th1/Th17 type cytokines, such as interleukin (IL)-2, IL-6, IL-23, IL-1, IL-8, IL-12, interferon (IFN)γ, and tumor necrosis factor (TNF)α, IL-17, and IL-21; meanwhile, protolerogenic Th2 cytokine subset production, i.e., IL-10 and IL-4 is simultaneously enhanced [18,29–33]. The concurrent downregulation of inflammatory cytokines and the upregulation of protolerogenic cytokines or neutraligands, as IL-10 and CCL22, with anti-inflammatory activity, allows T regulatory (Treg) cell expansion [30,32,33]. Furthermore, VDR agonists impair the inflammatory cascade in macrophages by targeting cyclooxygenase 2 (COX-2) and inducible nitric oxide synthase (iNOS), leading, in turn, to a significant reduction in nitric oxide (NO) and prostaglandin (PG)E2 content [29]. Vitamin D can suppress plasma-cell differentiation, B cell proliferation/differentiation, and antibody (IgG and IgM) production as well [34,35]. Importantly, DCs, macrophages, T cells, and B cells are able to synthesize vitamin D, which, in turn, downregulates the local expression of IL-12 and IL-23 and costimulatory molecules (i.e., CD40, CD80, CD86); therefore, modulating the immune response within the infiltrated target tissues as well [18,34]. Hence, an adequate level of vitamin D warrants a correct function of the immune response, promotes protolerogenic immune dominance, and protects from chronic inflammation, as shown in Figure 1.

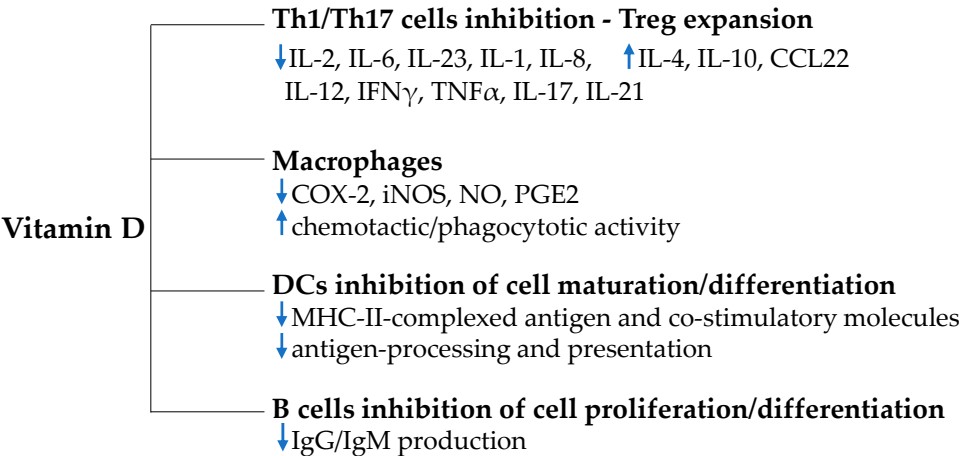

**Figure 1.** Vitamin D coordinates the anti-inflammatory immune response. Vitamin D exerts anti-inflammatory effects promoting protolerogenic immune dominance polarization.

At variance, vitamin D insufficiency/deficiency is shown to promote altered immune response and inflammation, which is shown to be associated with the development and maintenance of several pathologic conditions.

## 3. Vitamin D Orchestrates Skeletal Muscle Integrity

Skeletal muscle is a well-known target organ of vitamin D. In situ studies in muscle tissues documented the presence of VDR necessary for vitamin D uptake by myocytes, as confirmed in VDR knockout (VDRKO) mice and in humans carrying VDR natural mutations [36–39]. As from further evidence in humans and animals, vitamin D and VDR expression are necessary for muscle development, myocyte differentiation, muscular volume, and function maintenance, and physical performance [40–42]. Conversely, lower vitamin D levels and lower VDR expression associate with muscular deterioration by promoting biomolecular alterations, such as increases in oxidative stress and decline of antioxidant enzyme activity, which critically contribute to muscle atrophy [21]. Nevertheless, consistent results showing beneficial effects on the muscular mass and power after vitamin D treatment are missing and are still a matter of debate, as exhaustively summarized elsewhere [22]. In athletic and nonathletic populations, an adequate vitamin D level seems to ameliorate physical performance, likely due to hormone-induced enhancement of exercise capacity and cardiorespiratory fitness; in light of this observation, it would be conceivable to control the vitamin D status whenever physical activity is recommended as a therapeutic approach for CV risk prevention, primary or secondary to other diseases [43–46]. There is evidence that vitamin D accumulates in skeletal muscle cells, which can take up [25(OH)D] from the bloodstream through the vitamin D binding protein (DBP). Thus, it seems that the skeletal muscle provides a "functional store" for this hormone, which could be released upon a decrease in blood concentration. Of note, the ability of the skeletal muscle to maintain vitamin D circulating levels seems to be enhanced by regular physical exercise [47]. Isolated human skeletal muscle shows specific signals for the VDR protein with a different intensity expression before and after myotube fusion, although VDR presence in human skeletal muscle was previously debated [48,49]. In tissues, VDR is predominantly found in fast-twitch fibers, while slow muscles show lower receptor expression; regardless of muscle fiber types, a direct association between vitamin D level and intramyonuclear VDR concentration is documented [21]. Thus far, rather than hypovitaminosis D alone, the concomitant decline of circulating vitamin D and muscular VDR seems to be determinant for intracellular signaling damage and muscle function impairment. The very poor vitamin D/VDR status observed in age-related sarcopenia—a pathologic condition characterized by muscle mass reduction and function loss—or in myopathy associates with muscle decline and injury and very often contributes to the development of overlapping diseases, including dysmetabolism and diabetes [48,50–52]. Interestingly, human

skeletal muscle cells shortly exposed to a VDR agonist (in pulse/recovery assay) showed a rapid functional activation of the insulin (I)-dependent intracellular cascade, lasting for 12 h even in the absence of the ligand; this effect occurs simultaneously at glucose transporter type 4 (GLUT4) nuclear translocation, induced by non-genomic actions [48]. Hence, adequate vitamin D levels/VDR expression warrant muscular integrity and avoid musculoskeletal disorders in view of the multifaceted effects converging onto muscle development, mass/volume/performance maintenance, function regulation and metabolism control, therefore, largely contributing to general health homeostasis.

## 4. Skeletal Muscle-Immune System Interplay: A Two-Way Route

Nowadays, the link existing between skeletal muscle decline and impaired immune function is undeniable. But while several studies in the literature state the role of the immune system onto muscular function regulation, the concept of a reciprocal effect exerted by skeletal muscle onto immune regulation is quite more recent.

### 4.1. Immune Regulation of Skeletal Muscle Function

Muscle tissue is known to be under the control of the immune system. Skeletal muscle, like other tissues, retains a resident immune cell population, that could be further expanded by other infiltrating immune cells in case of pathologic events, such as an insult or injury, endotoxemia, inflammatory myopathies, obesity, diabetes. Indeed, the type and abundance of the resident immune cell population undergo modifications, depending on the muscle conditions and needs. Different types of immunocytes warrant skeletal muscle cell regenerative potential and homeostasis. I.e., immune cells following muscular tissue infiltration in response to micro- or macro injuries, eliminate necrotic cells and produce the biomolecules required for satellite cell proliferation/differentiation from the staminal niche [53]. It has been shown that CD4$^+$FoxP3$^+$ Treg type is the main T cell subset infiltrating the damaged muscle. The interplay satellite-Treg seems to be dependent on IL-33, present in the fibro/adipogenic progenitor-like cells within muscle tissue, and regulated by the protein amphiregulin, a growth factor directly promoting staminal cell growth [54,55]. Damaged Treg function, i.e., consequent to aging or diseases, promotes detrimental catabolic and inflammatory effects within skeletal muscle [53]. Those defects are reversed either by IL-33 injection, which rescued Treg function only in older mice, shifting the muscle transcriptome towards a reparative signature—the younger mice show a physiological ability to recover—or by polyclonal Treg injection, as documented by the significant improvement of disease activity in a rodent model of experimental autoimmune myositis (EAM) [54,56–59]. As for other tissue, during acute muscular damages, monocyte infiltration was associated with tissue TNFα positive gradient, following mast cell degranulation and neutrophil accumulation in the muscle. Within muscle tissue, monocytes differentiate into macrophages and polarize towards pro- or anti-inflammatory phenotypes. Indeed, in case of muscle injury or pathogens, the first inflammatory response, driven by monocyte to macrophage conversion, plays a pivotal role in activating pathogen clearance and muscle regeneration, orchestrating the participation of several other cell types, including T cells, mesenchymal stem cells, muscle satellite cells, myoblasts, and endothelial cells. Defects in macrophage activation/polarization, i.e., occurring during senescence, seem responsible for skeletal muscle inflammation development, and maintenance, as shown by studies in animals [60]. At variance, skeletal muscle inflammation, as observed in obesity, is associated with an increased number of skeletal muscle resident macrophages and contributes to insulin resistance (IR). A plethora of biomediators, mainly cytokines, chemokines, and membrane-bound factors, actively take part in those processes and are regulated by fine-tuned communications and interactions [61]. As an example, the drastic changes occurring in the immune system, consequent to aging or immunosenescence—a condition remodeling the immune system towards immune efficacy decline—significantly increase circulating pro-inflammatory molecules and markers, such as C protein reactive (CPR), TNFα, IL-6, released by different types of senescent cells, such as T lymphocytes, natural killer (NK), macrophages, neutrophils [62–64]. High blood levels of those mediators are considered a

mirror of chronic inflammatory status, which, in turn, is highly harmful to skeletal muscle integrity. As an example, patients affected by myositis show higher levels than healthy subjects of CXCL10, Th1 type chemokine known to trigger and maintain Th1-polarization within a vicious circle established between blood and local muscle cells, as addressed later in this text [48]. It has been documented that higher circulating pro-inflammatory cytokines likely act synergistically to trigger muscle wasting and even tissue loss, albeit some data discrepancy is reported, probably due to study setting and type, i.e., cross-sectional or longitudinal investigations, prospective cohort studies or metanalyses [64]. To date, whereas it is recognized that skeletal muscle health is highly affected by immune system status, evidence on skeletal muscle reciprocally influencing the immune system is more recent [65].

### 4.2. Skeletal Muscle Regulation of the Immune Function

For a long time, the immune system-muscle interplay has been thought to be unilateral. The concept that the skeletal muscle can act as a fine regulator of immune function is relatively new and enhances the complexity of this interplay. One of the first evidence comes from the observation that subjects with compromised skeletal muscle show an impaired immune response to pathogens, as clearly documented by the increased risk of infections in elderly subjects affected by sarcopenia. This condition, in fact, is associated with a higher risk of developing nosocomial infections or community-acquired pneumonia and mortality [66–68], besides disability and disability-related injuries (i.e., consequent to falls). Another interesting example comes from the observation of the higher risk of infection in healthy athletes undergoing skeletal muscle exhaustion for overtraining or muscular work overloading, i.e., elite athletes experiencing regular intense endurance training, as well summarized in dedicated reports [69–71]. Conversely, it has been suggested that balanced and well-scheduled physical activity could help in maintaining a correct immune response [70]. The concept that the muscle and the immune system could dialogue in a bidirectional way essentially relies on the renewed concept that the skeletal muscle does not serve only as pure locomotor organ, dedicated exclusively to sustain body and glucose homeostasis, but, remarkably, it behaves as a proper secretory organ with exquisite immune-regulatory and anti-inflammatory functions [72]. Paradoxically, this renewed concept mostly derives from studies aimed at verifying the muscular damage induced by the exercise itself, seen as potentially harmful a tissue stress factor. In fact, it is undeniable that high intensity or aerobic exercise can induce cellular and biomolecular modifications similar to a proper inflammatory response. However, all those "inflammatory-like" changes are transiently activated, shortly tapered-off after exercise, and occur in association with little immune cell infiltration [70]. In fact, regular exercising and gradual exercise adaptation induce the skeletal muscle to produce and release a plethora of immune-active factors with important beneficial regulatory functions, exerted both nearby and at a distance, as addressed in the following paragraph [72]. Indeed, myocytes could control and regulate immune function through cell-to-cell interactions (especially T cells-myocytes, as addressed above), membrane surface modulatory molecules present on skeletal muscle fibers, and soluble factors, in particular, cytokines, chemokines, and growth factors. Of note, skeletal muscle cells are, nowadays, regarded as non-professional APCs, able to present antigens through major histocompatibility complex (MHC) I and II, to produce immune biomolecules that, in turn, could shape the immune response. A detailed description of those processes is exhaustively reported in the literature [61,65]. This review would rather prefer to drive the attention on vitamin D interference at the signaling intersection between the immune system and skeletal muscle-derived cytokines (myokines).

## 5. Myokines, Immune Regulation, and Vitamin D: Converging Cell Signaling

Remarkably, upon contraction, the skeletal muscle releases a plethora of regulatory factors, named myokines, which are increasingly recognized to exert a fine-tuned control on biological age and health status, and are even proposed as frailty biomarkers [73]. To date, the proteomic profiling of the skeletal muscle secretome identified more than 300 myokines, potentially able to control nearby or distant organ function in autocrine, paracrine, and endocrine fashion [72,74,75]. Out of

all, some myokines, such as IL-6, IL-7, IL-15 seem to actively control and support immune system function [61]. IL-15-dependent signaling is engaged in human neutrophil migration and phagocytosis control, naïve T-cells survival, macrophage differentiation, and B cell proliferation [76]. Importantly, IL-15 helps the clearance of viral pathogens and tumor cells by promoting the proliferation and activation of NK cells and CD8+T cell homeostasis [77]. Accordingly, IL-15 knockout mice showed a defective immune response to viral vaccinia [78]. During skeletal muscle function deterioration, i.e., consequent to aging, diseases, or inactivity, a significant IL-15 decline is found in association with the immune defense decline and higher risk of infection [79]. IL-15 plays a pivotal role in macrophage differentiation—the cytokine itself can induce a cascade from alpha hydroxylase CYP27b1 activation necessary for [25(OH)D] conversion into bioactive [1,25(OH)$_2$D], ending in VDR activation and the induction of cathelicidin, an antimicrobial peptide largely engaged in first-line host defense [80,81]. A competent VDR is unquestionably required to mount a suitable immune response to intracellular pathogens [82]; the addition of vitamin D during IL-15-induced differentiation of monocytes into macrophages dose-dependently increased cathelicidin [83,84]. Interestingly, as from studies performed in human placental trophoblasts, a sex-dependent dimorphism in vitamin D metabolism would explain the major immune vulnerability to perinatal infections found in male fetuses and neonates vs. female ones. In fact, testosterone seems to decrease cathelicidin gene expression through CYP27B1 inhibition [85], thus impairing male immune defense. It would be very interesting to ensure further research in post-natal and adult life to investigate whether this mechanism might explain some sex-dependent differences (too often neglected) observed in response to infections [86], including the pandemic from the novel coronavirus disease 2019 (COVID-19) present at time of writing this text, which is reported to affect prevalently men [87]. IL-7, mostly produced in the thymus, is expressed and secreted by human skeletal muscle cells as well; it is classified as an exercise-regulated myokine, and exercise can restore IL-7 decline induced by aging [79,88]. IL-7-dependent signaling is critical for thymic development and function and thymic/extrathymic lymphocyte development and maturation (in particular, γδ T cells), necessary in maintaining immune competence, particularly, in first-line immune defense [89]. Lower circulating IL-7 is reported in conditions of impaired lymphopoiesis, as occurs in elderly humans or in subjects positive for human immunodeficiency virus (HIV) [90–94]. Studies on animals and humans include IL-7/vitamin D combination among the strategies to fight age-related immunosenescence, vulnerability to infections, and susceptibility to inflammatory diseases [95,96]. Furthermore, dysregulation of IL-7 signaling seems to be engaged in several autoimmune inflammatory processes [97]. In this context, vitamin D supplementation has been hypothesized as a tool to prevent autoimmune reactions by targeting and reducing IL-7 defective signaling [98]. Concerning IL-6, this is primarily known as a cytokine underlying inflammation-related diseases, including insulin resistance (IR) development [99,100]. Conversely, IL-6 exercise-induced pulsatile release controls myocyte metabolism, orchestrates myogenesis, and regulates inflammatory responses through the modulation of the IL-1 receptor antagonist (IL-1ra) or IL-10, both known to retain anti-inflammatory properties [101]. In addition, IL-6 can promote an anti-inflammatory profile in macrophages (M2-like), depending on the intracellular mediators involved in cytokine signaling, i.e., suppressor of cytokine signaling 3 (SOCS3) ablation can shift IL-6 from pro-inflammatory to anti-inflammatory mode [102,103]. I-like effects have been attributed to the myokine IL-6, since its transient increase in blood during muscular work—up to 100-fold, depending on exercise intensity/type—triggers hepatic glucose production and fat oxidation to provide substrate availability in response to fuel demand [104–106]. Summarizing, muscle-derived IL-6 could be identified as an energy level biosensor in conditions of energy shortage/demand, such as during physical exercise. Thus, the "good" IL-6 released by skeletal muscle cells upon muscular working shows important anti-inflammatory and metabolic properties, differently from the "bad" inflammatory IL-6, originating from immune cells or adipocytes. Interestingly, VDR agonists enhance the metabolic regulation exerted by the myokine IL-6, by improving glycemic control and I-sensitivity at the cellular level. In human skeletal muscle cells, the addition of a VRD agonist can significantly amplify protein release of IL-6 triggered by nutrient restriction, simultaneously promoting GLUT4

trafficking in lipid rafts—cell membrane areas dedicated to glucose internalization and involved in I sensitivity [48,107]. In vitamin D deficient trained male subjects, vitamin D intramuscular replacement determines improvements in metabolic function in association with a significant IL-6 increase 1 h post-resistance exercise [108], suggesting a recovery in skeletal muscle function after hormone supply. It is a matter of fact that promoting a good metabolic balance helps avoid homeostasis perturbance and inflammation. These beneficial effects of vitamin D combined with its other direct inhibitory actions onto inflammatory mediators so that this hormone sometimes is used as an adjuvant in the therapeutic approach to keep in control immune/inflammatory over-reactivity. For example, in rheumatoid arthritis (RA), an autoimmune disease characterized by higher circulating pro-inflammatory IL-6 and low vitamin D, the treatment with the IL-6 antibody tocilizumab retains higher efficacy in patients with sufficient hormone levels. Albeit the mechanism of function is still unclear, it has been hypothesized that vitamin D and IL-6 might act synergistically [109]. Table 1 summarizes the effects of vitamin D on IL-15-, IL7-, IL6-induced immune regulation.

**Table 1.** Vitamin D potentiates myokine-induced immune regulation.

| Myokines | Immune Regulation | Vitamin D |
|---|---|---|
| IL-15 | -macrophage differentiation<br>-NK cell activation<br>-CD8+T cell homeostasis<br>-neutrophils migration<br>-naïve T-cells survival<br>-B cell proliferation | -enhancement of IL-15-induced monocyte to macrophage conversion<br><br>-enhancement of anti-microbial activity |
| IL-7 | -thymic/extrathymic lymphocyte (γδ T cells) development and maturation<br>-immune competence<br>-first-line immune defense | -enhancement of IL-7 immune competence for protection against infections<br><br>-reduction of IL-7 defective signaling |
| IL-6 | -IL-1ra and IL-10 modulation<br>-M2-like macrophage profile<br>-anti-inflammatory/metabolic balance | -enhancement of anti-IL-6 antibody action (tocilizumab)<br><br>-improvement of IL-6-dependent metabolic/anti-inflammatory control |

The main immune-regulatory actions of skeletal muscle-derived IL-15, IL-7, and IL-6 are summarized together with the vitamin D effects on myokine-induced immune regulation. NK, natural killer; IL, interleukin; IL-1ra, IL-1 receptor antagonist.

Patients suffering from autoimmune idiopathic inflammatory myopathy (IM) show a marked decrease of VDR expression in muscle tissue, likely due to their poor vitamin D status, higher lipidemic/resistin profile, and, consequently, the potentially increased risk of developing overlapping metabolic diseases. These patients retain higher serum levels of CXCL10, a chemokine known to initiate/amplify Th1-polarization through a self-detrimental inflammatory loop established between systemic and local cell/tissue counterparts, as previously addressed [48]. Interestingly, a VDR agonist could reduce CXCL10 release by human skeletal muscle cells by 40% challenged by Th1 type inflammation, could prevent NF-kB activation downstream TNFα [110], and, importantly, can counteract class II HLA expression, necessary for immunogenic autoantigen presentation to CD4+T cells [111]. Considering that non-hypercalcemic VDR agonists could reduce the release of CXCL10 and other Th1/Th17 cytokines from CD4+T, shifting the immune profile towards Th2 subtypes [112,113], it is conceivable to envisage VDR agonists as immunoregulating therapeutic tools for the resolution of inflammation in myopathy, aging, or disease-related muscle decline [107,114–116]. However, although the supplementation/replacement of vitamin D status from deficient to normal potentially represents an optimal intervention to maintain human health, it is far from being conclusive, as addressed in the following paragraph.

## 6. Vitamin D Determination and Supplementation: Still Open Issues

The main source of vitamin D is sunlight exposition, followed by food intake. Aliments in nature, i.e., some vegetables and few animal-based foods, contain small vitamin D quantity, except for fish liver oils and fatty fish (salmon, mackerel, tuna) [117]; thus, fortified food has been introduced to maintain adequate hormone levels [118]. Since hypovitaminosis D seems diffused worldwide and tightly connected with poor general health status and higher mortality, vitamin D supplementation is considered a desirable intervention strategy for preventing non-skeletal chronic diseases [119–122]. However, vitamin D supplementation is a challenge;–the first question is related to the assessment of the levels to clearly define insufficiency/deficiency of vitamin D. Currently, values of [25(OH)D] (the more stable analyte in the blood, chosen for tbe level determination) ranging from 50–125 nmol/L seem sufficient to preserve overall health [123,124], 30 nmol/L is considered the threshold for hypovitaminosis D [22], while serum concentrations of more than 125 nmol/L are considered potentially toxic, due to side effects [125]. A further complication to accurately assess circulating vitamin D is represented by the high variability among available assays, methodologies, and analysis laboratories [126,127]. Besides the inter-variability, the intra-variability due to a lot of variation in reagents within a given methodology, contributes to the resulting discrepancy. Remarkably, the lack of standardized 25(OH)D data is a major contributor to the confusion existing on vitamin D status determination, consequently reflected in the uncertainty regarding the hormone supplementation approach [128], as exhaustively reported in the Consensus statement from the 2nd International Conference on Controversies in Vitamin D, recently published [22]. Furthermore, it is a matter of fact that life stage (infancy, adulthood, old age), lifestyle (indoor/outdoor, sedentary or training subjects, athletes), health status (health/disease, disease type), and physiological conditions (sex, fertile life, ethnicity) impact on the optimal serum concentrations of vitamin D, similarly to other hormone-related conditions [129–131]. In this scenario, the inconsistency between data from basic science, observational studies, and clinical trials is not surprising, albeit the pivotal role of vitamin D in health maintenance is clearly documented by both epidemiological and biomolecular studies. So far, well designed clinical trials taking into account both different populations and group-related specific variables should be strongly encouraged to clarify the causal relationships between vitamin D and non-skeletal human physiological and pathological conditions.

## 7. Conclusions

It is recognized that whole-body homeostasis and general health rely on the well-preserved function of the immune system and skeletal muscle, in association with vitamin D adequate levels. Indeed, the integrity of both these tissues widely contributes to and warrants a balanced immune dominance towards an anti-inflammatory status. Avoiding inflammation is considered the first step to counteract the development of a general pathologic status, as inflammatory load is known to be the common factor underlying different pathologies, regardless if disease type, all together often referred to as "diseasome". Accordingly, in the presence of muscle wasting and immune system dysfunction, progressive inflammatory and pathologic conditions develop and associate with poor outcomes, i.e., in metabolic and CV diseases, cancer, autoimmune diseases, just to mention some, and the list of the major diseases connected with skeletal muscle depletion and immune function deterioration is rapidly updated, as studies progress and clarify the underlying mechanisms. The concept that the immune system and skeletal muscle can reciprocally interact, modulating each other's function is quite recent; indeed, the interplay between those tissues has been long thought to be a one-way route, with the muscle being a "passive" target of the immune system. The renewed concept on skeletal muscle, nowadays considered a proper immune active secretory organ, led to reconsider the immune-muscle interplay as a dynamic two-way route. Within this integrated bidirectional dialogue, vitamin D, until now, is undeniably recognized as a fine regulator of the immune system and muscle development/maintenance, separately, which emerges to exhibit further multifaceted effects during the lifespan, interacting with different biological processes and converging intracellular signaling to protective actions, as reported above in this text and depicted in Figure 2.

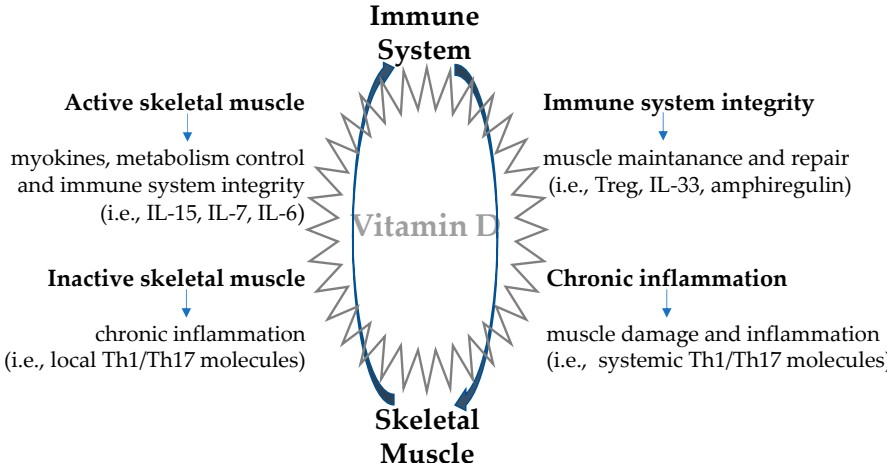

**Figure 2.** The three-side network relationship connecting the Immune System-Skeletal Muscle Network and vitamin D. Vitamin D acts within the immune-muscle axis, exerting regulatory protective effects.

In fact, depending on the microenvironment, vitamin D can behave as an exquisite inhibitory molecule, i.e., against inflammation, or a promoter/enhancer of homeostasis, i.e., in the regulation of metabolic processes. Hence, within the immune-muscle axis, adequate vitamin D levels can contribute to the careful orchestration of many biomolecular processes. At variance, hypovitaminosis D critically associates with immune system and skeletal muscle deterioration, inflammation, higher risk of disease development, and higher mortality. Whereas it is conceivable to include vitamin D supplementation in the therapeutic approaches to restore muscular and immune system function and counteract inflammation, still, clinical trials do not echo the promising results obtained by several epidemiological and molecular studies on this issue. This discrepancy likely depends on differences and bias in trial conditions. Further specific well-designed studies aimed to restrict the high variability are, therefore, encouraged. In particular, it is highly desirable and challenging that novel investigations would take into account the three-sided networking system connecting vitamin D effects, immune regulation, and skeletal muscle function as a multifaceted "holistic" system determinant for human health.

**Funding:** This research received no external funding.

**Acknowledgments:** Thanks to Tiziana Filardi, Sapienza University of Rome, for her kind assistance in reference preparation.

**Conflicts of Interest:** The author declares no conflict of interest.

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
