# Peer review of "Vitamin D Merging into Immune System-Skeletal Muscle Network: Effects on Human Health"

_applsci, doi:10.3390/app10165592_

Round 1
Reviewer 1 Report
It is interesting that the role that Vitamin D is an important immune system and the damage of the skeletal muscle is shown in this manuscript.
In addition, it is important for readers that immune-muscle interplay each other.
I think that it is easy to understand for readers that you illustrate 2 diagrams or tables as follows:
- The 3-side network connecting vitamin D effect, immune regulation and skeletal muscle function.
- Section 5 ‘Myokine, immune regulation and Vitamin D”
Author Response
I’m very grateful to this Reviewer for the suggestions.
- New Figure 2 depicts the three-side network relationship connecting Immune System-Skeletal Muscle Network and vitamin D.
- In Section 5, the new Table 1 summarizes the relationship Myokines, immune regulation and vitamin D.
Some changes have been carried out throughout the text and English has been reviewed by a native speaker.
Reviewer 2 Report
The review article provide the recent understanding of Vitamin D and the Immune System-Skeletal Muscle. However, the effects on Human Health are not very explored, especially in terms of disease.
General comments
- The Abstract need a sentence about the clinical importance of Vitamin D in the Immune System-Skeletal Muscle Network.
- In my opinion the review need figures, where the relationship between Immune System-Skeletal Muscle Network and the vitamin D
- The mechanism that Vitamin D coordinates anti-inflammatory immune response need a figure.
- The section Myokines, immune regulation and vitamin D: converging cell signaling need a table
- The conclusions are not clear, that are main effects of Vitamin D. What pathologies are associated with your needs? Pleased explain the sentence “Since hypovitaminosis D associates with immune system and skeletal muscle deterioration, higher risk of disease development and higher mortality, vitamin D supplementation seems a correct preventive/curative approach.?
Author Response
Thanks to this Reviewer for the constructive criticism and suggestions.
- A comment addressing the clinical importance of Vitamin D in the Immune System-Skeletal Muscle Network is now present in the Abstract, line 24-25; the Abstract has been modified accordingly in respect to word number limit.
- New Figure 2 depicts the three-side network relationship connecting Immune System-Skeletal Muscle Network and vitamin D.
- New Figure 1 depicts immune cell targeting by vitamin D promoting anti-inflammatory effects.
- Table 1 summarizes the section 5 Myokines, immune regulation and vitamin D: converging cell signaling.
- The paragraph “Conclusions” has been modified according to R2 requests. It has been stated that inflammation is associated with the development of different pathologies regardless disease type, as the inflammatory load represents the common triggering link. Albeit some diseases have been addressed in the text related to vitamin D-myokines-immune interplay (i.e., infections, metabolic and cardiovascular diseases, rheumatic diseases), the importance of skeletal muscle and immune system function integrity to prevent a general inflammatory status and, therefore, the risk of different pathologies, all together referred to as “diseasome”, has been highlighted in the revised manuscript. In particular, the following points have been added:
- “Indeed, the integrity of both these tissues widely contributes to and warrants a balanced immune dominance towards an anti-inflammatory status. Avoiding inflammation is considered the first step to counteract the development of a general pathologic status, as inflammatory load is known to be the common factor underlying different pathologies, regardless disease type, all together often referred to as “diseasome”. Accordingly, in presence of muscle wasting and immune system dysfunction, progressive inflammatory and pathologic conditions develop and associate with poor outcomes, i.e., in metabolic and CV diseases, cancer, autoimmune diseases, just to mention some. And the list of the major diseases connected with skeletal muscle depletion and immune function deterioration is rapidly updated, as studies progress and clarify the underlying mechanisms.” lines 360-369.
-“Within this integrated bidirectional dialogue, vitamin D, until now undeniably recognized as a fine regulator of immune system and muscle development/maintenance, separately, emerges to exhibit further multifaceted effects during all lifespan, interacting with different biological processes and converging intracellular signaling to protective actions, as reported above in this text and depicted in Figure 2.”, lines 374-378.
The sentence “Since hypovitaminosis D associates with immune system and skeletal muscle deterioration, higher risk of disease development and higher mortality, vitamin D supplementation seems a correct preventive/curative approach.” has been modified in “At variance, hypovitaminosis D critically associates with immune system and skeletal muscle deterioration, inflammation, higher risk of disease development and higher mortality. Whereas it is conceivable to include vitamin D supplementation in the therapeutic approaches to restore muscular and immune system function and counteract inflammation, still clinical trials do not echo the promising results obtained by several epidemiological and molecular studies on this issue.”, lines 386-391.
English has been reviewed by a native speaker.
Reviewer 3 Report
- Please modify “The first appearance of VDR biological activation by sub-nanomolar doses of the ligand is reported in the jawless fish lamprey about 550 million years ago [3].” This is not the exact meaning being conveyed by the article cited here.
- Line 47: The first studies evidenced a, please modify
- “Conversely, dysregulation/loss of immune and skeletal muscle functions, i.e., naturally occurring in aging, associates with immune system dysregulation, chronic inflammation and promotes overlapping disease development”- not clear, please modify.
- CCL22 upregulation along with T regulatory (Treg) cell expansion is allowed as well [30, 32, 33]. Please modify
- Importantly, DCs, macrophages, T cells, B cells are able to modulate immune response also within target tissues because of their ability to synthesize vitamin D, which, in turn, downregulates local expression of IL-12 and IL-23 and costimulatory molecules (i.e., CD40, CD80, CD86) [34]. Please modify, do the authors mean to say that these immune cells can synthesize vitD?
- What do the authors mean by low vitD levels are correlated with disease maintenance? Line 103
- Line 111: “lower vitamin D and VDR levels”, it should be lower vitD levels and lower VDR expression. Please modify
- “Conversely, lower vitamin D and VDR levels associate with muscular deterioration by promoting biomolecular alterations, such as increase of oxidative stress and decline of antioxidant enzyme activity, which critically contribute to muscle atrophy.” There are various studies suggesting no beneficial effects of vitD supplementation on muscle mass and power. The authors must mention both positive and negative findings and should discuss the possible reason behind this.
- Line 157-160: Please stress that this is the case with older mice only and younger mice have this ability.
- Line 166: are associated with a significant rise in blood of proinflammatory molecules and markers- please modify
- Line 188: in health athletes should be in healthy athletes
- Line 270-a synergistic effect vitamin D/IL-6 is hypothesized, please modify
- Line 280-release cells of CXCL10 and other Th1/Th17 cytokines- please check
- Since hypovitaminosis D seems worldwide diffused and tightly connected with general poor health status and higher mortality- not a complete sentence
- Please show the vitD-Muscle-myokine network and functional interrelationship by a line chart or a schematic diagram.
- As per the title of the review, the focus and discussion should be more in subsection 4.1
- Section 2: VitD is a well-known anti-inflammatory and immunomodulatory agent, the authors should focus discussion in this section related to muscle-vitD inflammatory and anti-inflammatory relationship rather than the other organs or systems
- Subsection 4.2 should be more elaborate and comprehensive
- The authors have discussed only CVS risk in correlation to vitD-muscle modulation, what is the effect on other systems? As the title is human health!
- Please check the grammar throughout the manuscript.
The manuscript describes the immune function of muscle and correlates with vitD suggestion muscle-vitD axis as a two-way immune axis. The description is well-formatted and described, however, at some places, it lost the context and went tangential. Please describe subsection 4.1 and 4.2 more comprehensively.
Author Response
I’m grateful to this Reviewer for the precious comments and suggestions.
- The sentence “The first appearance of VDR biological activation by sub-nanomolar doses of the ligand is reported in the jawless fish lamprey about 550 million years ago.” has been erased and reference number 3 is substituted with a new reference on vitamin D pleiotropic effects.
- The sentence “The first studies evidenced…” has been modified in “The strong association between vitamin D poor status and bone disorders or infections is well known.”, lines 46-47.
- The sentence “Conversely, dysregulation/loss of immune and skeletal muscle functions, i.e., naturally occurring in aging, associates with immune system dysregulation, chronic inflammation and promotes overlapping disease development” has been corrected, lines 60-63.
- The sentence “CCL22 upregulation along with T regulatory (Treg) cell expansion is allowed as well [30, 32, 33].” has been modified in “The concurrent downregulation of inflammatory cytokines and upregulation of protolerogenic cytokines or neutraligands, as IL-10 and CCL22, with anti-inflammatory activity, allows T regulatory (Treg) cell expansion [30, 32, 33].”, lines 91-94.
- The meaning is that those immune cell types can synthetize vitamin D and the sentence has been modified in “Importantly, DCs, macrophages, T cells, B cells are able to synthetize vitamin D, which, in turn, downregulates local expression of IL-12 and IL-23 and costimulatory molecules (i.e., CD40, CD80, CD86), therefore, modulating the immune response within infiltrated target tissues as well [18, 34], lines 98-101.
- The sentence has been modified in “At variance, vitamin D insufficiency/deficiency is shown to promote altered immune response and inflammation, which is shown to be associated with the development and maintenance of several pathologic conditions”, lines 107-109.
- “lower vitamin D and VDR levels” has been changed in “lower vitamin D levels and lower VDR expression”, line 117.
- The sentence addressing inconsistency on findings regarding vitamin D supplementation has been added: “Nevertheless, consistent results showing beneficial effects on muscular mass and power after vitamin D treatment are missing and still matter of debate, as exhaustively summarized elsewhere [22]”, lines 120-122.
- It has been specified that Treg rescue after IL-33 has been observed only in older mice as the “the younger mice show a physiological ability to recover”, lines 171-172.
- The sentence has been modified “…significantly increase circulating proinflammatory molecules and markers…”, line 189.
- “health” has been corrected in “healthy”, line 212.
- The sentence has been modified in “it has been hypothesized that vitamin D and IL-6 might act synergistically”, lines 304-305.
- The sentence has been corrected, line 320.
- The sentence has been completed, lines 333-334.
- New Figure 2 depicts the three-side network relationship connecting Immune System-Skeletal Muscle Network and vitamin D, and new Table 1 summarizes the relationship Myokines, immune regulation and vitamin D.
- The subsection 4.1 has been re-elaborated and enriched discussing the role of immune resident cells in muscle in phatophysiological conditions. In particular, the following sentences have been added: “Skeletal muscle, like other tissues, retains a resident immune cell population, that can be further expanded by other infiltrating immune cells in case of pathologic events, such as insult or injury, endotoxemia, inflammatory myopathies, obesity, diabetes. Indeed, the type and abundance of resident immune cell population undergo modifications, depending on muscle conditions and needs.”, lines 157-161; “As for other tissue, during acute muscular damages, monocyte infiltration is associated with tissue TNFα positive gradient, following mast cell degranulation and neutrophils accumulation in the muscle. Within muscle tissue, monocytes differentiate into macrophages and polarize towards pro- or anti-inflammatory phenotypes. Indeed, in case of muscle injury or pathogens, the first inflammatory response, driven by monocyte to macrophage conversion, plays a pivotal role to activate pathogen clearance and muscle regeneration, orchestrating the participation of several other cell types, including T cells, mesenchymal stem cells, muscle satellite cells, myoblasts and endothelial cells.”, lines 173-180. Concerning the suggestion to focus the discussion in this subsection, the author first wants to thank the Reviewer, as the subparagraph 4.1 introduces and describes the immune regulation of skeletal muscle function, which is undoubtfully a key point in this review. For a consistent architecture of the text, subsection 4.2 addressing the reciprocal skeletal muscle regulation of the immune function has been amplified as well, following point 18 request; then vitamin D action within immune-skeletal muscle axis is discussed in the dedicated section 5 (please, see also point 17).
- Section 2 of this review addresses and summarizes the main anti-inflammatory effects of vitamin D on immune system; the same is for Section 3, for vitamin D effects and skeletal muscle regulation. We agree with Reviewer that specific topics on vitamin D regulation of immune system are extensively present and exhaustively treated in literature; the same is for vitamin D effects on skeletal function. Undeniably, this review would rather “focus the attention on the immune system-skeletal muscle bidirectional crosstalk as a target point of vitamin D in health and disease” as stated in the aim in Introduction, and in Abstract, “This paper aims to focus onto immune system-skeletal muscle interplay as a multifaceted target of vitamin D in health and disease”. The choice to include brief separate overviews (sections 2 and 3) to recall separately vitamin D role onto immune and skeletal muscle regulation is for integrity and consistency of text architecture. In fact, in author’s opinion, a brief reminding of vitamin D distinct regulation of those two tissues would help to better introduce and discuss the vitamin D multifaceted protective roles exerted within immune-muscle axis, rather than on single tissue, as discussed in Section 5, as stated also in point 16. In this regard, Conclusions section has been modified with further final comments on the topic (lines 374-378).
- Subsection 4.2 has been re-elaborated and enriched with comments onto the immune-secretory activity of skeletal muscle, its inflammatory-like response to exercise, and skeletal muscle cell ability to behave as non-professional APC. In particular, the following sentences have been added: “Paradoxically, this renewed concept mostly derives from studies aimed to verify the muscular damage induced by the exercise itself, seen as a tissue stress factor potentially harmful. In fact, it is undeniable that high intensity or aerobic exercise can induce cellular and biomolecular modifications similarly to a proper inflammatory response. However, all those “inflammatory-like” changes are transiently activated, shortly tapered-off after exercise, and occur in association with little immune cell infiltration [70]. In fact, regular exercising and gradual exercise adaptation induce skeletal muscle to produce and release a plethora of immune-active factors with important beneficial regulatory functions, exerted both nearby and at distance, as addressed in the following paragraph [72].”, lines 219-227; “Of note, skeletal muscle cells are, nowadays, regarded to as non-professional APC, able to present antigens through MHC I and II, to produce immune biomolecules that, in turn, can shape immune response. A detailed description of those processes is exhaustively reported in literature [61, 65]. This review would rather prefer to drive the attention on vitamin D interference at signaling intersection between immune system and skeletal muscle-derived cytokines (myokines).”, lines 230-233; “At variance, skeletal muscle inflammation, as observed in obesity, is associated with an increased number of skeletal muscle resident macrophages and contributes to insulin resistance (IR).”, lines 182-184.
- Concerning vitamin D-muscle modulation, the relationship linking vitamin D status, muscular performance and CV system is addressed in the sentence, reported as follows, essentially to suggest that the inclusion of vitamin D level control may help in case physical activity is recommended for CV risk prevention, since adequate levels of the hormone seem to be benefical for muscular and cardiorespiratory function (“In athletic and nonathletic populations, an adequate vitamin D level seems to ameliorate physical performance, likely due to hormone-induced enhancement of exercise capacity and cardiorespiratory fitness; in light of this observation, it would be conceivable to control vitamin D status whenever physical activity is recommended as therapeutic approach for CV risk prevention, primary or secondary to other diseases [43-46].”). In paragraph 5 “Myokines, immune regulation and vitamin D: converging cell signaling” the text, indeed, addresses the beneficial effects of vitamin D-myokines-immune interplay on clearance of pathogens and tumor cells, infections (HIV), metabolic diseases (IR), rheumatic diseases (RA and IM). However, rather than single disease, this review would prefer to highlight the importance of an adequate vitamin D status necessary to avoid chronic inflammation and maintain general health. Accordingly, in the revised section “Conclusions”, additional comments on the role of vitamin D within immune-muscle axis to maintain general health have been addressed, highlighting the importance to avoid inflammation and, in turn, the development of a general pathologic status underlying several types of diseases, all together referred to as “diseasome” (please see the following text). “Indeed, the integrity of both these tissues widely contributes to and warrants a balanced immune dominance towards an anti-inflammatory status. Avoiding inflammation is considered the first step to counteract the development of general pathologic condition, as inflammatory load is known to be the common factor underlying different pathologies (regardless disease type), all together often referred to as “diseasome”. Accordingly, in presence of muscle wasting and immune system dysfunction, progressive inflammatory and pathologic conditions develop and associate with poor outcome, i.e., in metabolic and CV diseases, cancer, autoimmune diseases, just to mention some. And the list of the major diseases connected with skeletal muscle depletion and immune function deterioration is rapidly updated, as studies progress and clarify the underlying mechanisms.” lines 360-359.
- Grammar has been checked throughout the manuscript, English has been reviewed by a native speaker.
Overall, subsections 4.1 and 4.2 have been extended (please, see also responses at points 16 and 18), the text has been revised and integrated in order to ameliorate its consistency.
Round 2
Reviewer 2 Report
This article was revised appropriately.
I recommend accept.
Reviewer 3 Report
Thanks for addressing the concerns.